# Homogeneous Chronic Subdural Hematoma with Diverse Recurrent Possibilities

**DOI:** 10.3390/diagnostics12112695

**Published:** 2022-11-04

**Authors:** Woon-Man Kung, Yao-Chin Wang, Wei-Jung Chen, Muh-Shi Lin

**Affiliations:** 1Division of Neurosurgery, Department of Surgery, Taipei Tzu Chi Hospital, Buddhist Tzu Chi Medical Foundation, New Taipei City 23142, Taiwan; 2Department of Exercise and Health Promotion, College of Kinesiology and Health, Chinese Culture University, Taipei 11114, Taiwan; 3Department of Emergency, Min-Sheng General Hospital, Taoyuan 33044, Taiwan; 4Graduate Institute of Injury Prevention and Control, College of Public Health, Taipei Medical University, Taipei 11031, Taiwan; 5Department of Biotechnology and Animal Science, College of Bioresources, National Ilan University, Yilan 26047, Taiwan; 6Division of Neurosurgery, Department of Surgery, Kuang Tien General Hospital, Taichung 43303, Taiwan; 7Department of Biotechnology, College of Medical and Health Care, Hung Kuang University, Taichung 43302, Taiwan; 8Department of Health Business Administration, College of Medical and Health Care, Hung Kuang University, Taichung 43302, Taiwan

**Keywords:** hyperdense, hypodense homogeneous CSDHs, mean hematoma density (MHD), postoperative recurrence

## Abstract

**Background:** Evidence suggests that hyperdense (HD) chronic subdural hematomas (CSDHs) have a higher recurrence than hypodense (LD) chronic subdural hematomas. The value of mean hematoma density (MHD) has been proven to be associated with postoperative recurrence. The MHD levels in homogeneous CSDHs likely underestimate the risk of recurrence in HD homogeneous subtypes. **Methods:** This study investigated 42 consecutive CSDH cases between July 2010 and July 2014. The area of the hematoma was quantified to determine the MHD level using computer-based image analysis of preoperative brain CT scans. **Results:** In terms of the MHD distribution of the four types of CSDHs (homogeneous, laminar, separated, and trabecular), wide 95% CI (11.80–16.88) and high standard deviation (4.59) can be found in homogeneous types, reflecting a high variability in the MHD levels between cases (from low to high density). The categorization of homogeneous types into LD and HD (type five) displayed a minor standard deviation in the MHD levels for LD and HD subtypes (1.15, and 0.88, respectively). MHD values demonstrated concentrated distributions among the respective five types, compared to the four-type setting. **Conclusions:** In the current research, we provide a consideration that if LD and HD hematomas are separated from homogeneous CSDHs, the variability of the MHD quantification can potentially be reduced, thereby avoiding the possibility of undetected high-risk groups.

## 1. Introduction

For frontline physicians, specific characteristics of chronic subdural hematomas (CSDHs) deserve special attention. This disease commonly influences the elderly, who recently sustained a mild head injury. Specifically, the elderly population is susceptible to various health problems after the illness. Secondly, CSDHs, the so-called greater imitator, may appear in various forms of clinical manifestations and tend to be misdiagnosed due to vague symptoms. Thirdly, although the surgical technique is not difficult (creating burr holes in the skull to evacuate the hematoma), the postoperative recurrence rate has been reported to be as high as 23% [1,2,3,4]. Thus, qualitative and quantitative analyses have been broadly explored to investigate the mechanism underlying postoperative recurrence.

Nakaguchi et al. demonstrated four clinical types of CSDHs using preoperative CT features [5]. The pathogenesis course of CSDHs includes the initiation, development, maturity, and absorption of hematomas. The corresponding categories of Nakaguchi’s classification are: homogeneous (including hyperdense (HD), isodense, and hypodense (LD) as a reference to brain parenchymal density [6]), laminar, separated, and trabecular. This scheme has proven useful and has been widely cited [7,8]. However, accumulating evidence indicates that HD hematomas are extremely different from LD hematomas in postoperative recurrence. A high recurrence rate has been addressed in HD CSDHs [9,10,11,12], whereas the risk of postoperative recurrence has been shown to be low in LD CSDHs [8,13,14,15]. Because of the different bleeding tendencies for HD and LD hematomas, the combination of the two subtypes into one subject (homogeneous type) for analysis is likely to underestimate the risk of recurrence.

We previously demonstrated quantitative image analysis in terms of mean hematoma density (MHD) in CSDHs using preoperative CT images. MHD quantitation calculates an overall average density value of the entire subdural hematoma. A high MHD level reflects a greater extent of vascularity and hyperdense components within a hematoma. We statistically proved that MHD levels increase with the increasing risk of postoperative recurrence. Approximately, a one-unit increase in the MHD of a CSDH increases the odds of postoperative recurrence by a factor of 1.2 [16]. Moreover, we observed a linear correlation between MHD levels and the priority order of postoperative recurrence rate in Nakaguchi’s four types (Spearman’s rank correlation coefficient = 0.842, *p* < 0.001) [17]. In the current study, we aim to provide statistical evidence to verify the differences between homogeneous HD and LD CSDHs in terms of MHD quantification and recurrence risk. We aim to remind frontline physicians to avoid possible pitfalls in underestimating the recurrence rate of homogeneous HD CSDHs in patients.

## 2. Materials and Methods

### 2.1. Patients

This study reviewed the brain CT scans and medical reports of 42 consecutive patients who underwent surgery for CSDH at our institute between July 2010 and July 2014. Part of the information from this dataset of patients (35 patients, July 2010–July 2013) has been published [17]. However, a different clinical classification for patients and additional statistical analyses were conducted in the current study.

The 42 patients underwent burr-hole craniotomy drainage on one side (unilateral CSDH, n = 33) or both sides (bilateral CSDHs, n = 9). Only one burr hole with a closed-drainage system was made on each symptomatic side. To prevent confounding factors, such as different surgical approaches and a tendency to bleed, the following patients were excluded from the current study as our previous publication [16,17]: (1) Patients who had undergone craniotomy or the two-burr-hole technique on each symptomatic side; (2) patients with bilateral CSDHs including different Nakaguchi types; (3) patients that presented with coagulopathy due to liver disease or chronic renal failure; (4) patients with a homogeneous isodense hematoma, in which the hematoma boundary was difficult to distinguish from the brain parenchyma. This study was approved by the ethics committee of our institute.

After hematoma evacuation, intervals for follow-up CT scans were 24 h after surgery and at monthly intervals for at least 6 months until the recovery to a neurologically and radiologically stable status. If persistent neurological deterioration occurred, CT scanning was performed earlier.

For patients with a postoperative recurrence [18], a second operation was suggested due to persistent neurological symptoms and an increase in subdural collection with cerebral compression on the operated side, compared to the CT findings obtained 24 h after surgery [16].

### 2.2. Clinical Classification of CSDH

In the current study, preoperative cranial CTs were analyzed for all 42 CSDH patients. Cases were classified into the following types according to the internal architecture of CSDHs [5]: Homogeneous LD, homogeneous HD, laminar, separated, and trabecular.

### 2.3. Computer-Assisted Quantitative Analysis of CSDHs

The methodology for hematoma segmentation on brain CT scans is shown in Figure 1 of a previous report [19]. Briefly, the boundary of the hematoma was traced and outlined using image analysis software (GE PACS Web System). The density of the traced hematoma was calculated and presented in Hounsfield units (HU) for each axial slice.

Representative CT scans of CSDH patients with homogeneous LD and HD types, as well as the mode of tracing the clots on the images, are presented in Figure 1.

### 2.4. MHD Quantification for CSDHs [16]

The MHDs of unilateral CSDHs were calculated using the following equation: the mean of *Ai*, where *Ai* = HU of the traced hematoma for each axial CT slice (*i* = serial CT slice number). The MHDs of bilateral CSDHs were calculated using the equation: the mean of *Bi*, where *Bi* = the average HU of the traced hematoma on both sides of each axial CT slice (*i* = serial CT slice number).

### 2.5. Data Analysis

For continuous MHD data, we recorded the mean ± standard deviation (SD), median and interquartile range (IQR, the range between the 25th and 75th percentile), 95% confidence interval (CI), and coefficient of variation (CV). Differences in the MHD among groups were detected using the Kruskal–Wallis test and Dunn’s post-hoc test. A formal test for variance comparison was performed using the Fligner–Killeen test. Because of the small number in each group, age and follow-up duration were presented as the median (IQR) and performed using the Kruskal–Wallis test. Gender was presented as count and percentage and performed using Fisher’s exact test. Furthermore, multiple linear regressions were used to investigate the relationship between the MHD and CSDHs groups adjusted for age, sex, and follow-up duration. SPSS 22.0 (IBM Corp., Armonk, NY, USA) was used for statistical analyses. Fligner–Killeen tests were performed using R (version 4.2.1). The dot plots were drawn using MedCalc 11.5 (MedCalc Software, Ostend, Belgium). A two-tailed *p*-value of < 0.05 was considered to be statistically significant.

## 3. Results

### 3.1. Patient Characteristics

Forty-two patients, 32 males (76%) and 10 females (24%), with a mean age of 73.93 + 8.10 years (range 55–86 years) were included in the study between July 2010 and July 2014. These patients had a regular OPD follow-up return to the hospital until Oct 2016. The duration of the follow-up ranged from 30 to 72 months. The 42 cases were classified into four types according to the scheme of Nakaguchi [5,11] as follows: trabecular (n = 9), homogeneous (n = 15) [HD (n = 8), LD (n = 7)], laminar (n = 8), and separated (n = 10). The difference in age, sex, and follow-up duration among the 4 (or 5) groups were summarized in Appendix A. No significant difference was observed among these groups. Among the 42 patients, only three patients presented with a postoperative recurrence in the presence of neurological deficit and CT scan findings. Two were of the separated type and one was of a homogeneous HD subtype, as shown by the preoperative CT.

### 3.2. MHD Quantitations in Four Types

Previously, we revealed that higher MHD values were shown to be significantly related to postoperative recurrence [16], and significantly correlated with the probabilities of recurrence in Nakaguchi’s classification [17]. As shown in Table 1, the mean MHD for each CSDH type was: trabecular (12.43 ± 0.63 HU), homogeneous (14.34 ± 4.59 HU), laminar (26.07 ± 0.89 HU), and separated (36.82 ± 2.89 HU). The four types of CSDH according to their respective MHD levels—in ascending order of MHD level—are as follows: trabecular, homogeneous, laminar, and separate. Specifically, the postoperative recurrence rate of the four clinical types are discussed in the literature as follows: trabecular, homogenous, laminar, and separate (from lowest to highest risk) [5]. The sequence of the MHD levels in the current study completely correlated with the priority order of postoperative recurrence rate in Nakaguchi’s four clinical types, which suggests the reliability of the current dataset.

Significant differences in the MHD level were found among the four types (Table 1 and Figure 2, *p* < 0.001, Kruskal–Wallis test with Dunn’s post-hoc test). The significance of variation among the four groups was also observed (*p* < 0.001, Fligner–Killeen test). However, the difference in the MHD level between the trabecular and homogeneous types was not significant. In comparison with the four types, the widest 95% CI (11.80–16.88) and highest SD (4.59), IQR (9.00), and CV (32.00%) were exhibited by the homogeneous type (Table 1).

The MHD level for each patient is depicted in Figure 3. Differently from the other three types, the homogeneous MHD values were observed to be discrete in distribution. Therefore, we speculated that the variation in the MHD levels between the homogeneous (low to high) cases was so substantial that the statistical differences between the groups were not significant in the four-type classification.

### 3.3. MHD Quantitations in Five Types

For the next step, it was proposed that the HD and LD subtypes be separated out from the homogeneous type. The mean MHD for each subtype of CSDH type in the homogeneous group (14.34 ± 4.59 HU) was: HD (n = 8) (18.39 ± 0.88 HU), and LD (n = 7) (9.71 ± 1.15 HU) (Table 2). The MHD levels of the proposed five types were sorted in ascending order as follows: LD, trabecular, HD, laminar, and separate. Significant differences were found among the five groups in the five-type classification setting (Table 2 and Figure 4, *p* <0.001, Kruskal–Wallis test with Dunn’s post-hoc test). The significance of variation among the five groups was also observed (*p* < 0.001, Fligner–Killeen test).

Compared with the SD (4.59) before separation, a smaller SD was exhibited in the homogeneous hypodense (1.15) and hyperdense (0.88) hematomas (Table 2). The IQR (9.00 reduced to 1.72 and 0.51), 95% CI (11.80–16.88 reduced to 8.67–10.77 and 17.66–19.13), and CV (32.00% reduced to 11.85% and 4.78%) all exhibited similar trends. In contrast to the dispersed distribution of the homogeneous types in the four-type contexts, the MHD values were more concentrated close to the respective mean values of the five types (Figure 5).

We further analyzed the difference among these groups adjusted by age, sex, and follow-up duration using multiple linear regression (Table 3 and Table 4). The results were similar to Table 1 and Table 2. Among the four-type setting, there was no significant difference between the homogeneous and trabecular types (adjusted Beta = 1.90, *p* = 0.174), meanwhile, the laminar and separate types had a higher MHD than the trabecular type (adjusted Beta = 13.50 and 24.43, respectively, both *p* < 0.001). The results showed an increasing value in MHD from the homogeneous LD to the segregated type in the context of the five-type classification (compared to the trabecular type, adjusted Beta: 2.64 to 27.03, all *p* ≤ 0.004).

In terms of the MHD distribution, it was shown that the five types of CSDH were less dispersed in the subgroup than the four types of CSDH in the data distribution. Therefore, we surmised that the quantification of MHD under a less variable classification (five-type setting) could more accurately reflect the recurrence trend of the corresponding CSDH type.

## 4. Discussion

The following reports in the literature indicate that the postoperative recurrence rate associated with HD CSDHs was higher than that of LD CSDHs. Nomura et al. (41 patients, 1994) [20]; Oishi et al. (116 patients, 2001) [13]; Stanisic et al. (99 patients, 2005) [15]; Ko et al. (255 patients, 2008) [8]; Kong et al. (136 patients, 2012) [9]; and Stanisic et al. (107 patients, 2013, 2017) [11,12]. The reappearance of these conclusions highlights the observation that the HD CSDH has a higher recurrence rate than the LD hematoma. In summary of the abovementioned qualitative research, the following characteristics were involved in the formation of the HD CSDH, which contribute to the high risk of postoperative recurrence: The increase in vascularity [21], the proliferation of the outer membrane [22], a prominent neomembrane within the hematomas [5], high protein exudation from the outer membrane [23,24], and microbleeding into the hematomas [25].

CSDHs can exhibit different volumes of hematomas with high-density and low-density components in CT images. Among the hyperdense component, neovascular overgrowth into the CSDH membrane followed by repetitive microhemorrhages in the hematoma cavity can lead to hyperdense hematomas or high-density patterns of mixed hematomas [26]. The above phenomenon can be reflected in qualitative studies where the hyperdense component (hyperdense homogeneous and mixed densities) is shown to be a high recurrence factor, which contributes to retrospective studies of the etiology or prognostic assessment of the disease [6,27]. Nonetheless, our proposed MHD quantification research, counting the overall mean HU value beyond a certain site, is more useful in investigating the manifestation of high-density components throughout the entire hematoma cavity and the trend toward hematoma progression. As previously mentioned, the odds are 1.2 folds higher of the postoperative recurrence of CSDH per unit increase in MHD. Accordingly, we suggest that hematomas composed mainly of high-density components are more likely to have a postoperative recurrence because this type of hematoma tends to have a higher MHD. This proposed simple and practical computer-assisted quantitative assessment provides a platform for future studies. The quantitative MHD methods will enable a definitive analysis of the mean density of a CSDH, thus providing physicians with a comprehensive understanding of disease progression and leading to interventions to prevent a postoperative recurrence.

The CT features that were found in the abovementioned qualitative analysis, which related to postoperative high recurrence rate, e.g., abundant vascularity and neomembrane, can be clearly reflected in the computer-assisted quantitative method in terms of MHD. A correlation of recurrence risk with the MHD level was proved in our previous report [16]; i.e., CSDHs with a high MHD level are more likely to have a postoperative recurrence, such as the HD subtype. In the setting of the four-type classification, the HD and LD CSDHs are assigned the same type (homogeneous). Therefore, the MHD levels of the homogeneous type tend to be lowered because of the wide distribution of hematoma densities within this type. Specifically, a high MHD level in the HD subtype, which indicates high recurrence risk, can likely be overlooked if the MHD level of only the homogeneous type is calculated (the high MHD level of HD mixed with the low value of the LD subtype).

The homogeneous type includes the LD, isodense, and HD hematomas, exhibiting various hematoma densities in brain CT scans. Therefore, the MHD level in this group is very different, ranging from low to high levels. Analyzing the statistical information from our patients (four types), we found that the parameters of the homogeneous type were very different from the other three. Specifically, a 95% CI for the mean for each CSDH type was: trabecular (11.95–12.91), homogeneous (11.80–16.88), laminar (25.32–26.81), and separated (34.76–38.89). The standard deviation for each CSDH type was: 0.63 (trabecular), 4.59 (homogeneous), 0.89 (laminar), and 2.89 (separated). The coefficient of variation for each CSDH type was: trabecular (5.05%), homogeneous (32.00%), laminar (3.42%), and separated (7.84%). Investigating these parameters, a 95% CI for the mean in the homogeneous type has shown to be the widest of the four types. Not like the other three types with greater homogeneity, differences in the MHD levels between each case of homogeneous type were significant, i.e., among the four types, a high percentage of variation in the MHD level was observed in the homogeneous type. Based on these observations, we postulate that a large dispersion in the dataset of homogeneous type is likely associated with the low precision of MHD quantitation.

In the setting of the five-type classification, a 95% CI is as follows: LD (8.67–10.77), trabecular (11.95–12.91), HD (17.66–19.13), laminar (25.32–26.81), and separated (34.76–38.89). The standard deviation: LD (1.15), trabecular (0.63), HD (0.88), laminar (0.89), and separated (2.89). The coefficient of variation: LD (11.85%), trabecular (5.05%), LD (4.78%), laminar (3.42%), and separated (7.84%). Compared to the four-types, a relatively even distribution can be found for the MHD level in each type in the setting of the five-type classification, which explains our suggested five-type classification of CSDH patients.

CSDHs initiate from the cleavage of the dural border cell layer (as the dura-arachnoid interface) following a head injury [28]. The outer and inner membrane begins to form and encapsulate the hematoma cavity after approximately 1 to 3 weeks [29]. A variety of molecular pathways are likely involved in the progression of HD CSDHs, such as inflammatory cascade, exudation from the outer membrane, and repeated microhemorrhages from the neomembrane within the hematoma cavity [22,30,31]. Thus, the clinical course of the homogeneous stage progresses to the next stage (laminar type), since increased vascularity can be found along the inner membrane. This, in turn, means an increase in recurrence risk [5]. We postulate that LD CSDHs are associated with an earlier phase of the homogeneous stage [11], with less inflammatory activity and a low recurrence rate. HD CSDHs are related to the late phase of the homogeneous stage (prior to the laminar type), which underlie the inflammatory cycle and are associated with a high recurrence rate [14]. Our opinion is consistent with the reports of Stanisic et al. [11] and Edlmann et al. [14]. Because of the different pathogenesis courses of HD and LD CSDHs, we recommend separating out these two subtypes for analysis.

Several limitations exist in the current study. Only three patients had a postoperative recurrence (two separated types; one HD subtype). The number of recurrent patients is too small to make the current statistics conclusive in the correlations between recurrence and the classifications of CSDHs. Based on our previous experience in MHD analysis, we conducted the current study intending to statistically confirm that homogeneous HD and LD CSDHs are different categories in terms of MHD quantification. Furthermore, the MHD levels in the homogeneous LD subtype were the lowest among the proposed five types in the current dataset. Among the four types of CSDH in the literature, the trabecular type has the lowest recurrence rate [5]. At present, there is no relevant comparative analysis of recurrence rates between the homogeneous LD and trabecular types in the literature. A further quantitative investigation is warranted into the underlying mechanisms of CSDH recurrence.

In certain circumstances in organized CSDH with fibrous encapsulation and multiple internal septations, postoperative brain expansion can be compromised even after burr-hole evacuation. In this context, a craniotomy with extended membranectomy can be conducted as an effective strategy to promote brain re-expansion and prevent recurrence during initial surgical treatment or second-line treatment after failed borehole drainage [32,33]. As a further observation, a large craniotomy (about hematoma diameter ) with extended membranectomy reduced the reoperation rate compared to a small craniotomy (about 3–4 cm in diameter) with partial membranectomy [34]. However, this advanced surgical procedure may involve possible complications, such as seizure, or hemorrhage, especially in elderly patients [32,33,35,36], so special caution should be taken during the implementation. In our study, patients undergoing a craniotomy were excluded because we wanted to avoid possible confounding factors arising from the different surgical procedures. Other factors, including multiple surgical modalities, will be included in future studies to further investigate MHD-related postoperative recurrence.

## 5. Conclusions

Homogeneous HD CSDHs comprised primarily of HD components are more likely to present postoperative recurrence since this subtype has a higher MHD value compared to homogeneous LD hematomas; therefore, these patients deserve greater clinical attention, and it is recommended that those with a homogeneous LD hematoma be separated in the assessment of postoperative recurrence.

## Figures and Tables

**Figure 1 diagnostics-12-02695-f001:**
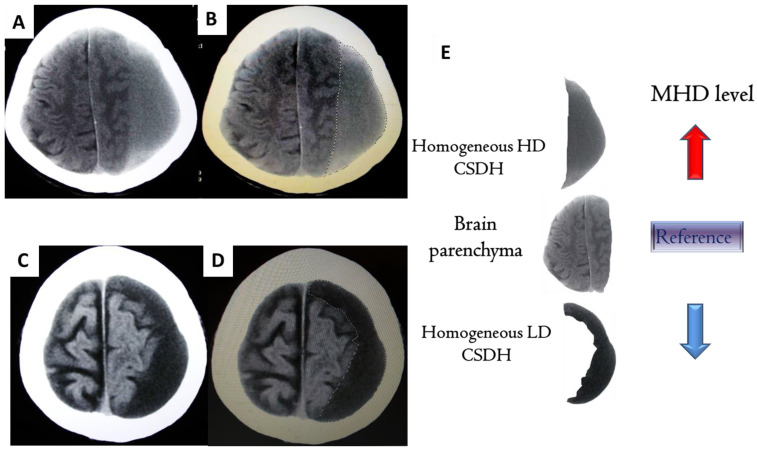
Representative CT scan of homogeneous hyperdense (HD) and hypodense (LD) chronic subdural hematoma (CSDH). Initial CT scans of patients with CSDH, homogeneous for HD (**A**) and LD (**C**) types showed that the density levels of the hematomas were higher and lower, respectively, than the brain parenchyma. As shown in (**B**,**D**), for each CT axial slice, the boundary of the CSDH is outlined under the PACS system and the density of the hematoma is computed and presented as Hounsfield units (HU). After determining the hematoma density for each axial section, the mean hematoma density (MHD) is calculated individually for each patient according to the formula. (**E**) Previous studies have shown that high values of MHD imply a high likelihood of hematoma instability, including hypervascularity, the production of a neomembrane, and even a high bleeding tendency. As shown in the schematic, although classified as homogeneous, the two types of hematomas, HD and LD, have vastly different MHD values.

**Figure 2 diagnostics-12-02695-f002:**
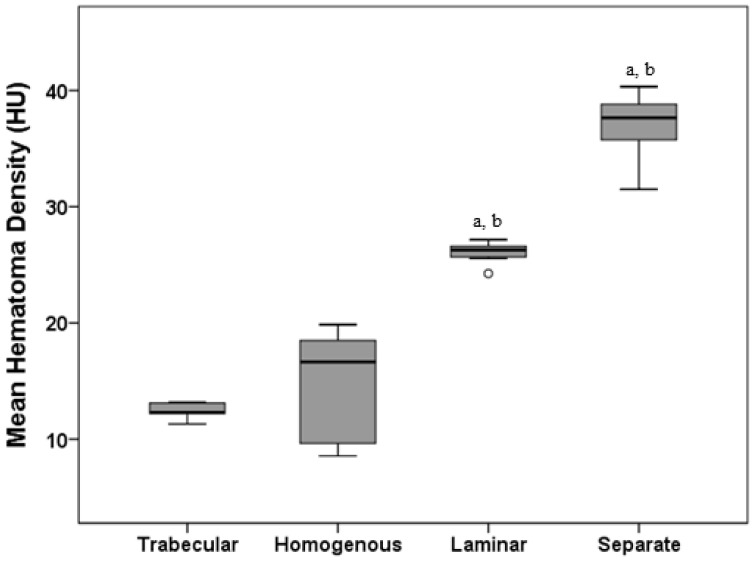
The distribution of MHD across four types of CSDHs based on internal structure. A significant difference was found using Dunn’s post-hoc test after Kruskal–Wallis test. ^a^ *p* < 0.05 compared to trabecular; ^b^ *p* < 0.05 compared to homogenous. Note: Values more than 1.5 IQRs but less than 3 IQRs from the end of the box are labeled as outliers (o).

**Figure 3 diagnostics-12-02695-f003:**
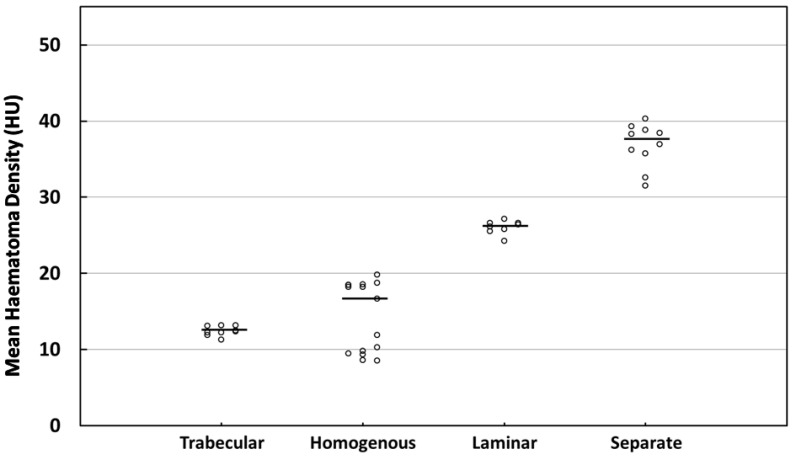
The distribution of MHD across four types of CSDHs based on internal structure. The dots indicated individual observations and the horizontal bar represents the median.

**Figure 4 diagnostics-12-02695-f004:**
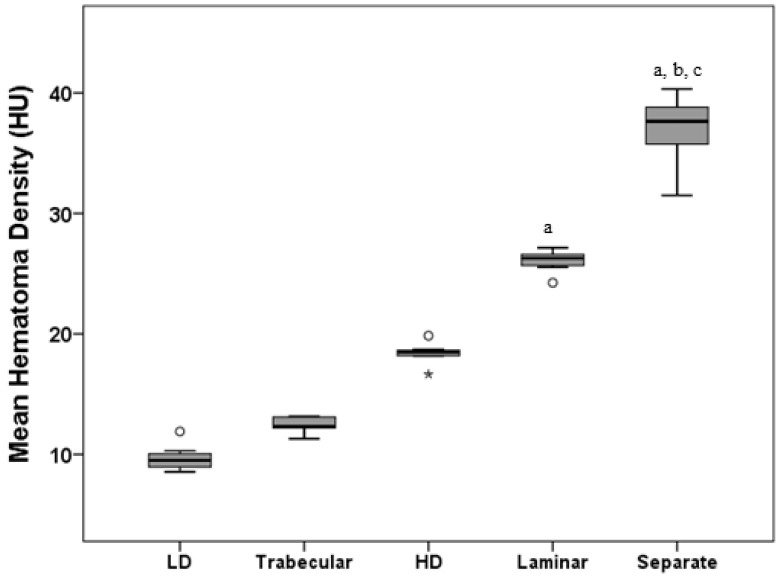
The distribution of MHD across five types of CSDHs based on internal structure. A significant difference was found using Dunn’s post-hoc test after a Kruskal–Wallis test. ^a^ *p* < 0.05 compared to LD; ^b^ *p* < 0.05 compared to trabecular; ^c^ *p* < 0.05 compared to HD. Note: Values more than three IQRs from the end of a box are labeled as extreme, denoted with an asterisk (*). Values more than 1.5 IQRs but less than 3 IQRs from the end of the box are labeled as outliers (o).

**Figure 5 diagnostics-12-02695-f005:**
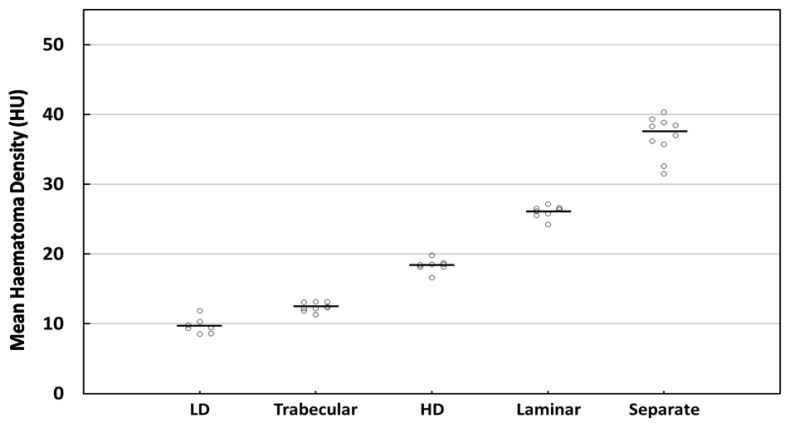
The distribution of MHD across five types of CSDHs based on internal structure. The dots indicated individual observations and the horizontal bar represents the median.

**Table 1 diagnostics-12-02695-t001:** The relationship between MHD and type of CSDHs based on internal structure (four-type classification).

	Trabecular	Homogenous	Laminar	Separate	*p*-Value ^a^	*p*-Value ^b^
N	9	15	8	10	<0.001	<0.001
Mean ± SD	12.43 ± 0.63	14.34 ± 4.59	26.07 ± 0.89	36.82 ± 2.89		
Median (IQR)	12.33 (1.07)	16.65 (9.00)	26.28 (1.01)	37.65 (3.98)		
95% CI	11.95–12.91	11.80–16.88	25.32–26.81	34.76–38.89		
CV	5.05%	32.00%	3.42%	7.84%		

SD: Standard deviation; IQR: Interquartile range; CI: Confidence interval; CV: Coefficient of variation. ^a^ *p*-value of the Kruskal–Wallis test. ^b^ *p*-value of the Fligner–Killeen test for homogeneity of variances.

**Table 2 diagnostics-12-02695-t002:** The relationship between MHD and the type of CSDHs based on internal structure (five-type classification).

	Homogeneous LD	Trabecular	Homogeneous HD	Laminar	Separate	*p*-Value ^a^	*p*-Value ^b^
N	7	9	8	8	10	<0.001	0.011
Mean ± SD	9.71 ± 1.15	12.43 ± 0.63	18.39 ± 0.88	26.07 ± 0.89	36.82 ± 2.89		
Median (IQR)	9.50 (1.72)	12.33 (1.07)	18.50 (0.51)	26.28 (1.01)	37.65 (3.98)		
95% CI	8.67–10.77	11.95–12.91	17.66–19.13	25.32–26.81	34.76–38.89		
CV	11.85%	5.05%	4.78%	3.42%	7.84%		

SD: Standard deviation; IQR: Interquartile range; CI: Confidence interval; CV: Coefficient of variation. ^a^ *p*-value of the Kruskal–Wallis test. ^b^ *p*-value of the Fligner–Killeen test for homogeneity of variances.

**Table 3 diagnostics-12-02695-t003:** The MHD among 4 types of CSDHs adjusted for age, sex, and follow-up duration.

	Beta	SE	*p*-Value
CSDHs			
Trabecular	reference		
Homogenous	1.90	1.37	0.174
Laminar	13.50	1.57	<0.001
Separate	24.43	1.48	<0.001
Age, years	0.060	0.060	0.346
Gender (Male vs. Female)	−0.670	1.190	0.579
Follow-up duration, month	−0.010	0.040	0.751

**Table 4 diagnostics-12-02695-t004:** The MHD among 5 types of CSDHs adjusted for age, sex, and follow-up duration.

	Beta	SE	*p*-Value
CSDHs			
Homogeneous LD	reference		
Trabecular	2.64	0.84	0.004
Homogeneous HD	8.60	0.88	<0.001
Laminar	16.20	0.88	<0.001
Separate	27.03	0.83	<0.001
Age, years	0.02	0.03	0.539
Gender (Male vs. Female)	−0.10	0.62	0.867
Follow-up duration, month	−0.02	0.02	0.467

## Data Availability

The data used to support the findings of this study are included within the article.

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
