# Peer review of "Homogeneous Chronic Subdural Hematoma with Diverse Recurrent Possibilities"

_diagnostics, 2022, doi:10.3390/diagnostics12112695_

Round 1
Reviewer 1 Report
Manuscript ID: 1890256
Title: Homogeneous chronic subdural hematoma with diverse recurrent possibilities
Authors: Woon-Man Kung et. al.
Journal: diagnostics
Statistical Review
1) In Tables 1 & 2, instead of presenting (Q1, Q3), it would more appropriate to present IQR
2) The SNK post-hoc test compares average hematoma density levels across the 4 (or 5) type classification, adjusting for multiplicity. A significant p-value is expected as an outcome of this comparison, as other literature resources suggest. But, the authors real concern is whether or not differences in variation exist among the 4 (or 5) groups. A formal test for variance comparison is Bartlett’s test (under normality assumption) or Fligner’s test (as a non parametric alternative). In any case, the authors can easily test normality using a typical procedure (e.g. Shapiro’s test) and then choose the variance test accordingly. The p-value of the variances comparison should be in included in Tables 1 & 2.
3) Adjustment in comparing the mean or the variance of hematoma density levels for other information (e.g. age, sex, follow-up duration e.t.c.), should be performed and presented, even unofficially. The limited sample may prevent the derivation of safe and sound conclusions, but still an idea is formed on how this adjustment alters author’s findings.
4) It is suggested figures 2 & 4 to be replaced with box-plots
Date: 30 September 2022
Author Response
The point-to-point response letter is attached

Reviewer 2 Report
Dear Author,
This is an interesting study; however, studies have reported the density of cSDH and outcomes in detail
Multiple Densities of the Chronic Subdural Hematoma in CT Scans - PMC (nih.gov)
https://www.ncbi.nlm.nih.gov/pmc/articles/PMC3772285/
Radiological prognostic factors of chronic subdural hematoma recurrence: a systematic review and meta-analysis | SpringerLink
https://pubmed.ncbi.nlm.nih.gov/33094383/
Author Response

(The authors gave the same response as above.)

Reviewer 3 Report
The Authors conducted a study on a series of 42 patients affected by CSDH, focusing the analysis on hight MDH level that reflects greater extent of vascularity and hyperdense components within the heamptoma. They observed that MDH level increased with the increasing risk of recurrence. The Authors didd a good job and conclusions and limitations are correctly stated.
There are some comments that I belive could strengthen the paper:
1) there are several articles regarding the tratment of Organized Chronic Subdural Hematomas treated by cranitomy and extensive membranectomy in order to avoid the recurrence. This topic should be discussed.
2) Not clear appear the criteria of exclusion of the study: craniotomy, duble burr holes etc.
3)Reference 34 is repeated twice 41.
Author Response

(The authors gave the same response as above.)

Round 2
Reviewer 2 Report
The authors have made the suggested changes